# An acute dose of intranasal oxytocin rapidly increases maternal communication and maintains maternal care in primiparous postpartum California mice

Caleigh D. Guoynes *, Catherine A. Marler

Department of Psychology, University of Wisconsin, Madison, WI, United States America

* guoynes@wisc.edu

**Data Availability Statement:** All relevant data are within the manuscript and its Supporting Information files.

## Abstract

Maternal-offspring communication and care are essential for offspring survival. Oxytocin (OXT) is known for its role in initiation of maternal care, but whether OXT can rapidly influence maternal behavior or ultrasonic vocalizations (USVs; above 50 kHz) has not been examined. To test for rapid effects of OXT, California mouse mothers were administered an acute intranasal (IN) dose of OXT (0.8 IU/kg) or saline followed by a separation test with three phases: habituation with pups in a new testing chamber, separation via a wire mesh, and finally reunion with pups. We measured maternal care, maternal USVs, and pup USVs. In mothers, we primarily observed simple sweep USVs, a short downward sweeping call around 50 kHz, and in pups we only observed pup whines, a long call with multiple harmonics ranging from 20 kHz to 50 kHz. We found that IN OXT rapidly and selectively enhanced the normal increase in maternal simple sweep USVs when mothers had physical access to pups (habituation and reunion), but not when mothers were physically separated from pups. Frequency of mothers' and pups' USVs were correlated upon reunion, but IN OXT did not influence this correlation. Finally, mothers given IN OXT showed more efficient pup retrieval/carrying and greater total maternal care upon reunion. Behavioral changes were specific to maternal behaviors (e.g. retrievals) as mothers given IN OXT did not differ from controls in stress-related behaviors (e.g. freezing). Overall, these findings highlight the rapid effects and context-dependent effect a single treatment with IN OXT has on both maternal USV production and offspring care.

## Introduction

Quality of maternal care has significant impacts on offspring survival outcomes across many mammalian species [1–5]. These studies underscore the importance of maternal behavior from an evolutionary perspective. However, the proximate mechanisms that reinforce maternal care remain more elusive. Several studies in rodents illustrate that pup whines, high energy calls produced by pups, quickly and reliably elicit maternal care [6–11]. Other studies,

**Funding:** C.M. 1355163 National Science Foundation https://www.nsf.gov/awardsearch/showAward?AWD_ID=1355163 The funders had no role in study design, data collection and analysis, decision to publish, or preparation of the manuscript.

**Competing interests:** The authors have declared that no competing interests exist.

however, show that mothers are more apt to exhibit care in response to stressful events or disturbances and that pup calls do not influence their care above and beyond the disturbance [12]. It has also been shown that female house mice prefer calls from their own pups and can locate their own faster than alien pups [13]. These studies show that pup whines can elicit changes in maternal behavior. However, the role of maternal vocalizations in this relationship has not been studied. Adult rats, including mothers, make spontaneous vocalizations that typically occur above 50 kHz when presented with drug and social [14, 15]. The association of reward with these calls in rats is interpreted as an indicator of a positive internal state. In mothers, USVs may indicate maternal motivation but could also reinforce maternal care and the mother-offspring bond [16, 17] or reduce maternal anxiety [18].

Complementing the stimulus of maternal and pup vocalizations, the neuropeptide hormone oxytocin (OXT) plays an important role in processing and producing behaviors that support maternal care. OXT modulates many social behaviors including bonding and parental care [19–23]. During mammalian birth, OXT increases to stimulate parturition and milk let-down in mothers; this increase was likely co-opted during evolution to also facilitate maternal care [24–27]. Immediately after parturition, rising levels of peripheral estrogen [28] prime the neural substrates that respond to OXT to initiate maternal behaviors in rats [26]. Additionally, OXT knockout mice have greater latency to onset of maternal behaviors [29]. In the brain, OXT antagonists blocked maternal behavior after natural delivery in pregnant rats [30]. This reveals that central OXT is important for initiating maternal care in rodents. Acute activation of maternal care by OXT is indirectly supported by an optogenetics study in which dopamine neurons were activated in the anteroventral periventricular nucleus (AVPV) that monosynaptically connects to and activates OXT neurons in the paraventricular nucleus (PVN), resulting in enhanced maternal care [31]. OXT receptor densities are also important. In rats genetically selected for differences in maternal care, high grooming compared to low grooming females had more OXT receptors in the bed nucleus of the stria terminalis, medial preoptic area, central nucleus of the amygdala, and these differences were observed in maternally-experienced females that were either non-lactating and lactating [32, 33]. Collectively, these studies provide strong evidence that OXT plays an important role in activating and coordinating maternal care.

OXT also plays a role in the production and perception of vocalizations. In mice and other rodents, the majority of vocalizations occur above 20 kHz and are called ultrasonic vocalizations (USVs) [34–37]. In OXT knockout mice, OXT null pups emit fewer USVs in response to separation from their mother compared to wildtype mice [38]. This suggests that OXT may enhance pup communication with their mother. OXT can also improve the signal-to-noise ratio in mothers responding to pup calls via mediation of temporal inhibition and excitation in the left auditory cortex of female mice, leading to increased pup retrievals [39]. These data provide evidence that OXT is changing the perception and social salience of pup calls, leading to increased maternal care. Furthermore, in humans, the OXT receptor has a polymorphism (rs53576) with functional significance. The genotype GG (presumably produces more OXTRs compared to AG or AA genotypes) is associated with better ability to discriminate content of language under noisy conditions [40]. This suggests that across taxa, OXT may play an important neuromodulatory role in promoting sensory processing and behavioral response to social auditory information.

A key social behavior that has previously not been measured is maternal vocalizations. We speculated that mothers modulate vocalization quantity or type when interacting with their offspring and that maternal vocalizations would be associated with maternal care. Moreover, we predicted that OXT would modulating these vocalizations.

The California mouse (*Peromyscus californicus*) is a strictly monogamous, biparental rodent species well-suited to examine how OXT modulates auditory sensory processing, vocal

production, and social behavior. California mice have a diverse, well-characterized repertoire of ultrasonic vocalizations (USVs) including simple sweeps, complex sweeps, syllable vocalizations, barks, and pup whines [41–44]. Previous recordings between mothers and pups indicated that the primary call types from mothers were maternal simple sweeps and the primary call type from pups were pup whines. While this suggests that these two call types are important in mother-pup contexts, both maternal simple sweeps and pup whines have also been recorded in other social contexts [43, 45, 46].

In the current study, we aimed to address the gaps in our understanding of proximate mechanisms that contribute to maternal care by determining the association between maternal vocalizations and maternal care and whether an acute dose of IN OXT in primiparous female California mice could rapidly increase both maternal vocalizations and care. Previous studies have shown that IN OXT alters behavior within five minutes of administration [47] and can have behavioral effects that can persist for 30–50 min after administration [48]. As we were not manipulating the OXT system in the pups, we did not expect to see an effect of OXT on pup whine USVs. We hypothesized that 1) maternal care would be associated with maternal USVs and that 2) IN OXT would have a positive effect on maternal care. Specifically, we predicted that during our behavioral paradigm, IN OXT would increase maternal care, increase maternal USV production, and enhance the correlation between pup USVs and maternal USVs and that physical separation would disrupt the pro-social effects of IN OXT.

## Methods

### Animals

University of Wisconsin-Madison Institutional Animal Care and Use Committee approved this research. We used 24 primiparous postpartum female *P. californicus* aged 5–10 months because of ages of previously unpaired females available within our colony. Females across this age range also show equivalent corticosterone responses to corticotrophin releasing hormone and dexamethasone challenge [49], suggesting that females within this age range have comparable glucocorticoid responsiveness. Females were pair-housed (1 female, 1 male, and 1–3 pups per cage; 48 × 27 × 16 cm) under a 14L: 10D light cycle. Animals were maintained in accordance with the National Institute of Health Guide for the Care and Use of Laboratory Animals. Female and male mice were randomly paired to an unrelated mouse and were housed in their home cage. After females were visibly pregnant, cages were checked once daily for pups and gave birth in the home cages. Mothers were randomly assigned to either the saline control group (N = 11) or the IN OXT group (N = 13). The mode of pups per litter was two, and there was a range of pups per litter (1–3). Number of pups was considered for use as a covariate, but in the statistical models, including this variable a) did not explain additional variance and b) reduced the power of the statistical comparison. Pup number across treatments was very similar—average number of pups for mothers in the saline condition was 2.11, and average number of pups for mothers in the OXT condition was 1.91.

### Intranasal oxytocin preparation

Mothers were infused intranasally with either sterile saline control or IN OXT (0.8 IU/kg) (Bachem, Torrance, California). The IN OXT dose was equivalent to doses used in other rodent species [50–52] and similar to weight-adjusted doses used in clinical studies examining the effects of IN OXT on social deficits in autism [47]. IN OXT was dissolved in saline and prepared in one batch that was aliquoted into small plastic tubes and frozen at 20˚C. IN OXT was defrosted just prior to administration. A blunt cannula needle (33-gauge, 2.8 mm length; Plastics One, Roanoke, Virginia) was attached to cannula tubing, flushed, and filled with the

compound, then attached to an airtight Hamilton syringe (Bachem, Torrance, California). The animal was scruffed and 25 uL of compound was expelled dropwise through the cannula needle, alternating between left and right nostrils for each mouse. Rodents are obligate nose-breathers and thus the solution was quickly absorbed into the nasal mucosa (~10seconds). One person conducted all IN OXT administrations throughout the entire procedure to maintain consistency in handling and IN OXT infusion.

We chose to use the method of intranasal administration of IN OXT for two primary reasons. (1) IN OXT is used in clinical studies and is less invasive, does not require special transporters for the molecule, and is presumed to be less stressful compared to intracerebroventricular [53]. (2) IN OXT shows similar behavioral effects as centrally administered OXT, increases CSF and plasma concentrations of OXT, and reaches the relevant brain areas in both humans and animal models [54–58]. Several studies have also shown changes in plasma OXT concentrations that peak between 15 to 30 min post-administration [59, 60]. These results suggest IN OXT passes through the blood-brain barrier to exert central effects with minimal stress on the animal. In California mice, behavioral effects of IN OXT are consistent with the outcomes of central OXT manipulations suggesting that IN OXT is reaching the brain [61, 62]. Other studies indicate that some of the effects of IN OXT are acting through peripheral mechanisms [63–65]. Regardless of whether IN OXT is directly targeting the brain, is acting through peripheral mechanisms, or a combination of both, IN OXT has been shown to rapidly alter social behavior in adult California mice [66].

## Behavioral testing and USV recording

In order to test the effects of IN OXT on acute maternal care, we conducted this experiment in a novel recording chamber where mothers and pups could be briefly isolated from the father. Separation from the mate may be a mild stressor but would occur in natural populations in response to competing demands such as pup care, foraging and defending territories. Moreover, California mice do not exhibit a change in short- or long-term paternal care in response to corticosterone [67], show limited correlations between individual baseline corticosterone levels and behavior [68], parents exhibit blunted behavioral response to predator odor stress [69], and diel corticosterone cycle between single mothers and paired mothers does not differ [70]. While we cannot exclude an effect of a baseline level of stress, it is both a normal experience for these mice in the wild, and we do not expect corticosterone to influence the results of our current experiment above and beyond our IN OXT manipulation.

On postnatal day (PND) two to three, fathers were temporarily removed from the home cage, and the home cage with the mother and her pups was transferred to a behavioral testing room. In the behavioral testing room, mothers were randomly selected to receive 25 microliters of either 0.8 IU/kg IN OXT or saline control intranasally. Immediately after dosing, mothers and pups were placed into one side of a partitioned two-chambered apparatus (45.0 cm × 30.0 cm × 30.0 cm) that contained a circular opening (3.8 cm in diameter, center of opening 7 cm from the side wall) covered by a wire mesh (**Fig 1A**). This apparatus, like their home cages, had approximately 1/2 inch of aspen shavings covering the entire floor. Microphones sensitive to ultrasonic frequencies (described below) were placed on each side of the divider, such that the microphones were far enough apart to identify the source (chamber) of the calls [71] (**Fig 1A**).

For each test, there were always two researchers present who coordinated activation of the audio software and the video camera at the same time using visual cues. This coordination allowed us to subsequently compare USVs and behavior with temporal precision. The test consisted of three phases that occurred in immediate succession: habituation,

**(A)**

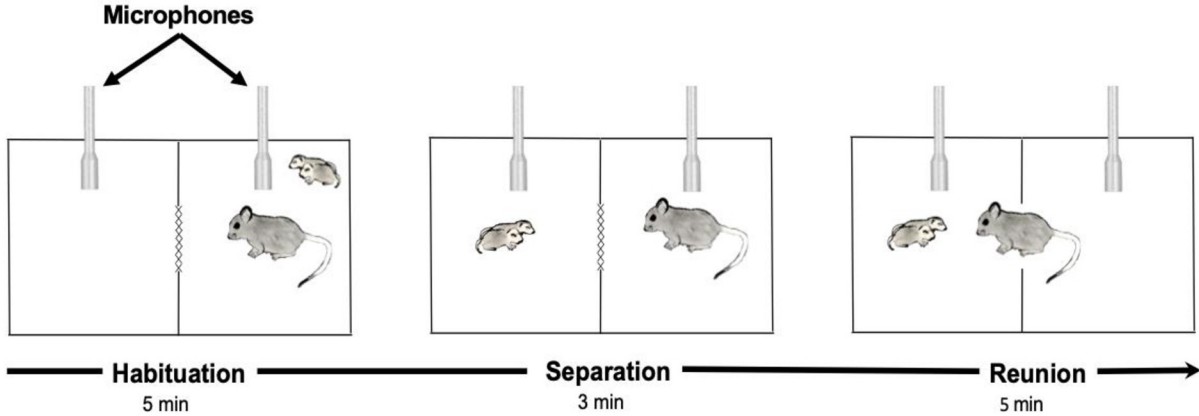

**(B)**

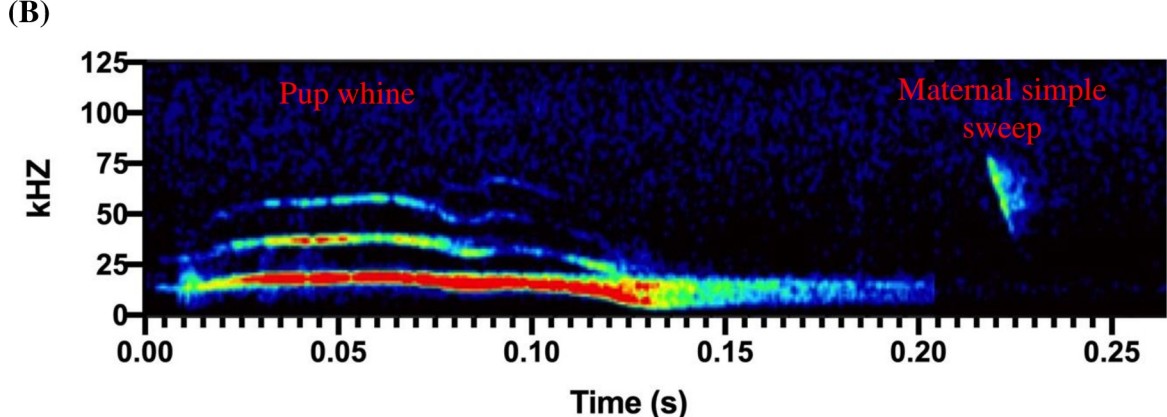

**Fig 1. (A)** Schematic for experimental design. Mother and pup (PND 2–3) groups were temporarily removed from their home cage and placed in the right side of two-chambered apparatus (five min) to habituate to the testing arena. Next, pups were moved from the right chamber and placed into the left chamber (three min). Lastly, the researchers lifted the mesh gate separating the right and left chambers, allowing mother-pup interactions (five min). Animals not to scale in diagram. **(B)** Ultrasonic vocalizations (USVs) on a spectrogram. Pup whines have multiple harmonics, a peak frequency around 20 kHz, and downward modulation at the end of the call that distinguish these calls from adult syllable vocalizations. Maternal simple sweeps have short downward-sweeping vocalizations that sweep through multiple frequencies, typically between 80 kHz and 40 kHz.

separation, and reunion. During the five-min habituation phase mothers and pups were placed together on the right side of the chamber with the partition down and allowed to freely interact with each other. During the three-minute separation phase, the pups were removed from the mother and placed on the other side of the partitioned divider. Mothers remained in the right-most chamber and pups were isolated in the left-most chamber. This setup allowed visual, auditory, and olfactory communication between pups and their mother, but restricted physical contact between individuals until the mesh wire was removed. Lastly, during the five-minute reunion phase, the mesh divider was lifted, and mothers could retrieve and interact with their pups. USVs and video were recorded for the entire 13-minute period. We chose a five-minute initial mother-pup interaction time to allow mothers time to adjust to the chamber and to mirror the time in the reunion

phase where we measured latency to enter the chamber. This time period is important because it is the first time that the mothers and her pups are removed from the home cage and the father, so this time period served as an initial measure of behavior. Results from a pilot study measuring maternal retrievals indicates that five-minutes allowed most mothers to enter the chamber, approach the pups, and engage in maternal behaviors. We shortened the separation phase to three minutes because it was still sufficient to see signs of maternal distress but minimized the time that the pups were away from their mother.

### Behavior quantification

All behavioral videos were scored in random order and by an independent observer blind to treatment. During each video, maternal behaviors (licking and grooming, huddling, and retrieving/carrying) were quantified. Of note, unlike house mice and rats that show several different types of nursing and huddling behavior, California mouse mothers do not show arched back nursing [72, 73]. The definitions of behaviors measured are detailed in an ethogram (**S1 Fig**). To gain insights into the correlations between maternal behavior and maternal and pup USVs, videos were coded by the exact time that mothers were huddling, licking and grooming, and retrieving/carrying. Huddling was counted when mothers were physically over their pups' bodies [74]. Retrieving/carrying was counted when mothers picked up their pups and transferred them to a different location. Using the precise times that mothers engaged in different types of maternal care (or none at all) throughout the 13-minute testing window, we counted the maternal USVs within those windows. This allowed us to determine how maternal behavior was related to maternal USV production.

### Ultrasonic vocalization analysis

Techniques used for recording were similar to those previously used in our laboratory [44, 46]. USVs were collected using two Emkay/Knowles FG series microphones capable of detecting broadband sound (10–120 kHz). Microphones were placed at the far ends of each of the two chambers. Microphone channels were calibrated to equal gain (− 60 dB noise floor). We used RECORDER software (Avisoft Bioacoustics) to produce triggered WAV file recordings (each with a duration of 0.5 s) upon the onset of a sound event that surpassed a set threshold of 5% energy change [36]. Recordings were collected at a 250 kHz sampling rate with a 16-bit resolution. Spectrograms were produced with a 512 FFT (Fast Fourier Transform) using Avisoft-SASLab Pro sound analysis software (Avisoft Bioacoustics). The only USVs found in these recordings were pup whines and maternal simple sweeps. Pup whines have a peak frequency around 20 kHz [75, 76] and the typical downward modulation at the end of the call often distinguishes these calls from adult syllable vocalizations (Nathaniel Rieger, Jose Hernandez, & Catherine Marler, unpublished) (**Fig 1B**). The lower frequencies in the pup whine can also be heard by human ears (below the ultrasonic range). Maternal simple sweeps were categorized by short downward-sweeping vocalizations that sweep through multiple frequencies, typically between 80 kHz and 40 kHz [77] (**Fig 1B**). It is extremely rare for pups to produce simple sweep USVs during PND 0–4 (Rieger, N. S., Hernandez, J. B., and Marler, C. M., unpublished). When young pups do produce simple sweeps, they are produced much quicker, and present completely vertical on the spectrogram [75]. This makes these rare pup simple sweeps easy to distinguish from the slower adult simple sweep USVs (**Fig 1B**). Because of their different spectrogram and acoustic properties, all USVs could be categorized and counted by combined visual and auditory inspections of the WAV files (sampling rate reduced to 11,025 kHz, corresponding to 4% of real-time playback speed).

## Data analysis

Statistical analyses were conducted using the program R. Significance level was set at p<0.05 for all analyses and all tests were two-tailed. All reported p-values were corrected using Benja-mini-Hochberg false discovery rate corrections to control for multiple comparisons when effect of an X variable was tested for a relationship with multiple Y variables.. Grubb's outlier test was performed, and outliers for maternal vocalizations and freezing were removed from all analyses. Two mice (one control and one IN OXT-treated mouse) were Grubb's outliers (*p<0.05)* for freezing (likely due to train noise during the test). One control mouse was a Grubb's outlier (*p<0.05)* for both maternal and pup USVs. All analyses used a generalized lin-ear mixed model (GLMM). Treatment condition was used in all models as each female was given one treatment (between-subjects design). Thus, when a relationship between two vari-ables was significant, treatment was left as a moderator in the model even if not significant.

**Effects of IN OXT on vocalizations.** To assess differences in total USV production across testing conditions, a within-subjects two-way ANOVA was used. To correct for differences in total time of the three testing phases, average number of USVs/second during each phase for each animal was calculated and used to compare the five-minute versus three-minute trials. To assess main effects of treatment, Student's t-test was used in each of the three time points.

To examine the effects of IN OXT in the relationships between maternal simple sweep USVs and pup whine USVs, an interactive multivariate model was used (e.g. `[Maternal behavior] ~ [Maternal USV] + [treatment]`). Factor included in all models was treatment condition.

**Effects of IN OXT on maternal and non-maternal behaviors.** For maternal and non-maternal behavioral analysis, behavioral changes after the separation event, were calculated to examine reunion behaviors with and without OXT administration. To examine changes in behavior over time and after the separation challenge with and without OXT, the scores from total duration of each behavior in the reunion phase were subtracted by total duration of each behavior in the habituation phase (Reunion-Habituation). Thus, positive scores indicate more of the behavior was observed during the reunion phase and negative scores indicate more of the behavior was observed during the habituation phase. To compare main effects of IN OXT and saline control on maternal behavior, Students t-tests were used to assess behavioral out-comes (**Fig 3**). To calculate total maternal behavior, amount of time spent huddling and amount of time spent retrieving were summed. To calculate total non-maternal behavior, amount of time spent autogrooming, rearing, and freezing were summed.

**Correlations between maternal care and maternal USVs and maternal care and pup USVs.** To assess for mediation by IN OXT in the relationships between (a) maternal USVs and maternal behavior and (b) maternal behavior and pup USVs, a multivariate comparison was used. Factors included in the model were treatment condition and the interaction between treatment and maternal behavior.

## Results

### Effects of IN OXT on vocalizations

To determine how testing conditions affected vocal production in mothers and pups, we first assessed number of maternal and pup USVs per second across the habituation, separation, and reunion phases, and in response to IN OXT versus saline. Controlling for within subject analy-ses and treatment effects, mothers made fewer simple sweeps/second during the separation phase compared to the habituation or reunion phases ($F_{2,20}$ = 13.00, *p*<0.00001). (**Fig 2A**). In the habituation phase, IN OXT mothers produced more simple sweeps than control mothers

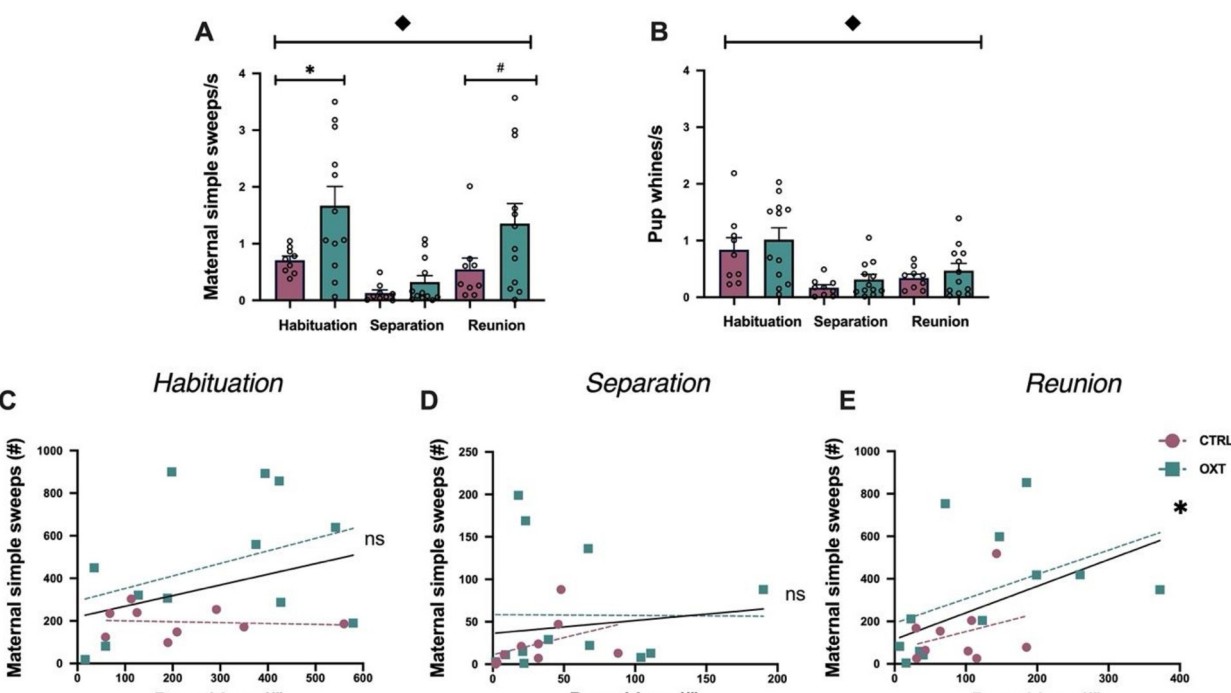

**Fig 2. Rapid effects of IN OXT on USV production in mothers and pups. (A)** All mothers made more simple sweeps when given free access to their pups (during the habituation and reunion phases). Importantly, IN OXT mothers made more simple sweeps when given free access to their pups during habituation and reunion, but not when they were physically apart from pups during separation. **(B)** All pups made more whines when first placed into the chamber during the habituation phase. There were no effects of maternal IN OXT on pup USVs. Pups made more whines during the habituation phase than the separation and reunion phases. There was no effect of IN OXT treatment on pup whines during habituation, separation or reunion. Mediation analysis of the relationship between maternal USVs, pup USVs and treatment in **(C)** the habituation phase showed no simple effects of maternal USVs, no simple effects of treatment, but IN OXT showed a nonsignificant trend for the two-way interaction between maternal simple sweep USVs and treatment. **(D)** The separation phase showed no simple effects of maternal USVs, no simple effects of treatment, and no effect of interaction. **(E)** The reunion phase showed a significant positive correlation between maternal simple sweeps and pup whines, no simple effect of treatment, and no effect of interaction. Correlation line in black is the average slope across treatment groups; magenta and teal lines are the slopes for the saline and OXT treatments, respectively. ♦ p<0.05 for differences across time conditions; *p<0.05 for differences between control and OXT; #p<0.10 for differences between control and OXT.

($F_{2,20}$ = 5.83, $p$<0.03, $\Delta R^2$ = 0.23) (**Fig 2A**). In the separation phase, IN OXT and control mothers did not differ in number of simple sweeps produced ($F_{2,20}$ = 1.86, $p$ = 0.19, $\Delta R^2$ = 0.09) (**Fig 2A**). Similar to the habituation phase, in the reunion phase, IN OXT mothers showed a nonsignificant trend for producing more simple sweeps than control mothers ($F_{2,20}$ = 3.13, $p$ = 0.08, $\Delta R^2$ = 0.15) (**Fig 2A**).

Across the three phases, pup USVs showed no effect of IN OXT but did show changes in vocal production frequency across the testing phases. Controlling for within-subject analyses, pups made more whines during the habituation phase than the separation and reunion phases ($F_{2,20}$ = 25.26, $p$<0.0000001) (**Fig 2B**). Pups with IN OXT and control mothers did not differ in number of USVs produced in the habituation phase ($F_{1,20}$ = 0.35, $p$ = 0.56, $\Delta R^2$ = 0.02), separation phase ($F_{1,20}$ = 1.64, $p$ = 0.22, $\Delta R^2$ = 0.08) or the reunion phase ($F_{1,20}$ = 0.68, $p$ = 0.42, $\Delta R^2$ = 0.04) (**Fig 2B**).

Next, we examined the relationship between number of maternal simple sweeps and pup whines using a model with treatment as a covariate. There were no significant correlations between maternal and pup USVs in either the habituation ($F_{1,20}$ = 1.63, $p$ = 0.22) (**Fig 2C**) or the separation phase ($F_{1,20}$ = 0.03, $p$ = 0.86) (**Fig 2D**). However, during the reunion phase, maternal simple sweeps and pup whines positively correlated ($F_{1,20}$ = 7.51, $p$<0.02, $\Delta R^2$ =

0.21). For each pup whine, there were approximately 1.17 maternal simple sweeps (**Fig 2E**). There were no simple effects of OXT on the correlation between maternal-pup USVs: habituation ($F_{1,20}$ = 0.25, $p$ = 0.62) (**Fig 2C**), separation ($F_{1,20}$ = 1.27, $p$ = 0.27) (**Fig 2D**), reunion ($F_{1,20}$ = 0.15, $p$ = 0.69) (**Fig 2E**). Finally, our model also assessed the two-way interaction between maternal simple sweep USVs and treatment. IN OXT showed a nonsignificant trend for the two-way interaction between maternal simple sweep USVs and treatment, with IN OXT animals having a more positive slope (non-significant trend) than controls ($F_{1,20}$ = 3.71, $p$ = 0.069) (**Fig 2C**). Slope for IN OXT-treated mothers did not differ from controls in either the separation ($F_{1,20}$ = 0.07, $p$ = 0.79) (**Fig 2D**) or reunion ($F_{1,20}$ = 2.90, $p$ = 0.10) (**Fig 2E**) phases.

### Effects of IN OXT on maternal and non-maternal behaviors

To assess the impact of IN OXT on maternal care following separation from pups, we assessed latency to enter the chamber with pups and then calculated change in maternal care from habituation to reunion to measure changes in other types of maternal behavior. This allowed us to determine how IN OXT affected response to pup separation above and beyond any initial effects of IN OXT observed in the habituation phase. In the beginning of the reunion phase, IN OXT mothers showed a non-significant trend for a shorter latency to approach pups ($F_{1,20}$ = 3.63, $p$ = 0.10, $\Delta R^2$ = 0.16) (**Fig 3A**). We also tested for main effects of IN OXT in several maternal and non-maternal behaviors. Negative scores mean that the behavior occurred more frequently during habituation and positive scores mean that the behavior occurred more frequently during reunion. Mothers tended to retrieve/carry more during the habituation phase and huddle more during the reunion phase. Mothers given IN OXT did not show any differences in huddling from controls ($F_{1,20}$ = 0.62, $p$ = 0.44) (**Fig 3B**). However, mothers given IN OXT showed a significantly more positive change in retrieval/carrying behavior from the

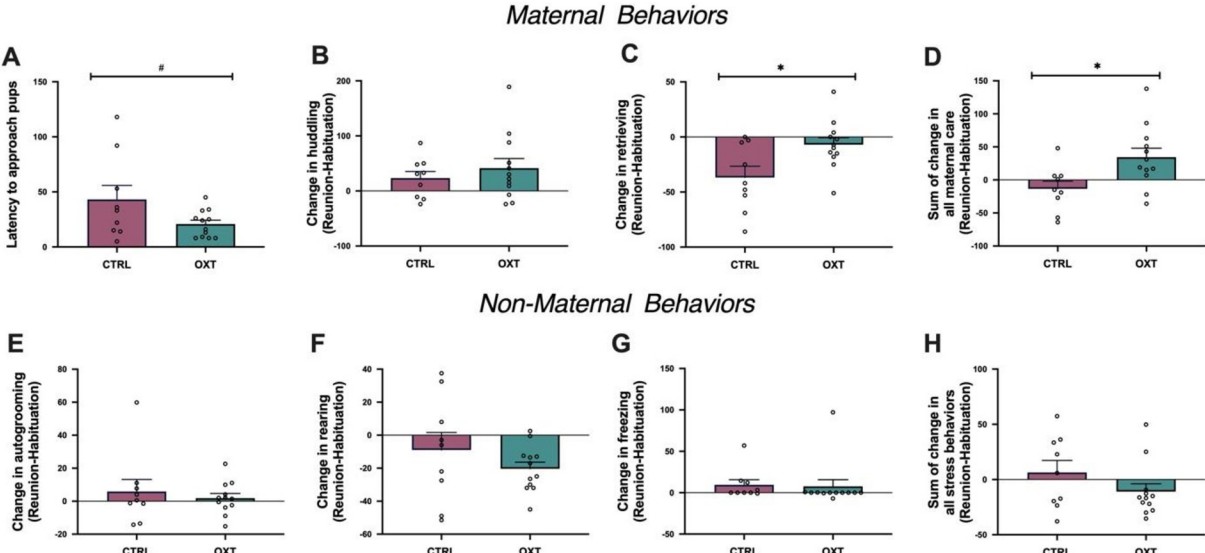

**Fig 3. Rapid effects of OXT on change in maternal and non-maternal behavior from habituation to reunion.** *Maternal behaviors*: **(A)** There was a non-significant trend for mothers given IN OXT to have a shorter pup approach latency. **(B)** There were no treatment differences for maternal huddling. **(C)** Mothers given IN OXT showed a significantly greater decrease in retrieval/carrying behavior from the habituation to the reunion phase. **(D)** IN OXT had a net positive effect on total maternal care relative to controls from the habituation to reunion phases. *Non-maternal behaviors*: **(E)** There were no treatment effects for change in autogrooming, **(F)** rearing, or **(G)** freezing. **(H)** There was no net change in non-maternal behaviors from the habituation to reunion. Correlation line collapsing across treatment groups. *$p < 0.05$; #$p < 0.10$.

habituation to the reunion phase ($F_{1,20} = 6.71$, $p<0.05$, $\Delta R^2 = 0.26$) (**Fig 3C**). This is driven by high rates of retrieval in control mothers during habituation. High rates of retrieval are associated with less efficient maternal care in mother rats in home and novel environments [78–80] and virgin and mother mice in novel environments [81, 82]. Thus, the IN OXT mothers are likely more efficient at maintaining offspring care during this disruption. While mothers from both groups increased huddling behavior post-separation from pups, control mothers decreased retrieval/carrying behavior while IN OXT mothers maintained a consistent level of retrieval/carrying behavior. The net increase in maternal care for IN OXT mothers from habituation to reunion ($F_{1,20} = 6.6$, $p<0.02$, $\Delta R^2 = 0.26$) (**Fig 3D**) is therefore largely due to changes in retrieval behavior.

In order to determine if IN OXT was acting on neural systems that were specific to maternal care, we also tested for main effects of IN OXT on measures of activity (autogrooming, rearing) and stress/anxiety (freezing). From habituation to reunion, there were treatment differences in autogrooming ($F_{1,20} = 0.32$, $p = 0.58$) (**Fig 3E**), rearing ($F_{1,20} = 1.23$, $p = 0.28$) (**Fig 3F**), or freezing ($F_{1,20} = 0.03$, $p = 0.85$) (**Fig 3G**). When summing all activity and stress-related behaviors, there was no net effect on non-maternal behaviors from habituation to reunion ($F_{1,20} = 1.94$, $p = 0.18$) (**Fig 3H**). In this context, IN OXT is not influencing general, non-maternal behaviors in response to an offspring separation event. The individual means and SEMs for each behavior in each phase are also reported in **S1 Table**.

## Correlations of maternal care with maternal USVs and pup USVs

To determine whether maternal simple sweeps were associated with specific maternal and investigative behaviors during each of the three testing phases, we correlated number of maternal USVs, which were always simple sweeps, during each phase with the corresponding behavior while controlling for moderation by IN OXT administration. In the habituation phase, maternal simple sweeps positively correlated with licking behavior ($F_{2,20} = 12.04$, $p<0.007$, $\Delta R^2 = 0.34$) (**Fig 4B**). There was also a nonsignificant trend for a correlation between maternal simple sweeps and huddling ($F_{2,20} = 4.48$, $p = 0.072$, $\Delta R^2 = 0.18$) (**Fig 4A**), and no effect associated with retrieving/carrying ($F_{2,20} = 0.44$, $p = 0.51$, $\Delta R^2 = 0.02$) (**Fig 4C**). There was also, however, a significant moderation by IN OXT in the relationship between maternal retrievals and maternal simple sweeps. During habituation, mothers given IN OXT carried/retrieved pups less than mothers given saline ($F_{2,20} = 9.95$, $p<0.005$, $\Delta R^2 = 0.33$) (**Fig 4C**) but note that overall, IN OXT mothers had consistent retrieval levels across the test (**Fig 2B**). In the separation phase, there was no correlation between time the mother spent at the mesh divider and maternal simple sweep USVs ($F_{2,20} = 1.2$, $p = 0.58$, $\Delta R^2 = 0.058$) (**Fig 4D**). Other maternal behaviors could not be assessed because of the mesh divider between mothers and pups. In the reunion phase, maternal simple sweeps positively correlated with maternal retrievals/carrying ($F_{1,20} = 7.65$, $p<0.037$, $\Delta R^2 = 0.26$) (**Fig 4G**). Other maternal behaviors did not correlate with maternal simple sweeps in this phase: huddling ($F_{1,20} = 0.48$, $p = 0.99$, $\Delta R^2 = 0.02$) (**Fig 4E**) and licking ($F_{1,20} = 0.001$, $p = 0.98$, $\Delta R^2 = 0.00$) (**Fig 4F**). Thus, overall, there were associations between maternal simple sweeps and maternal care, but the maternal behavior that correlated with maternal simple sweeps varied depending on context.

Next, we correlated pup whine USVs with specific types of maternal behavior to see if these pup calls were associated with a specific maternal response. During the habituation phase, there was a significant positive correlation between pup whines and maternal huddling ($F_{1,20} = 8.93$, $p<0.024$, $\Delta R^2 = 0.30$) (**Fig 5A**), but not with maternal licking ($F_{1,20} = 3.26$, $p = 0.174$, $\Delta R^2 = 0.13$) (**Fig 5B**) or retrieval/carrying behavior ($F_{1,20} = 1.14$, $p = 0.26$, $\Delta R^2 = 0.04$) (**Fig 5C**). During the separation phase, pup whines did not correlate with time that the mother spent at

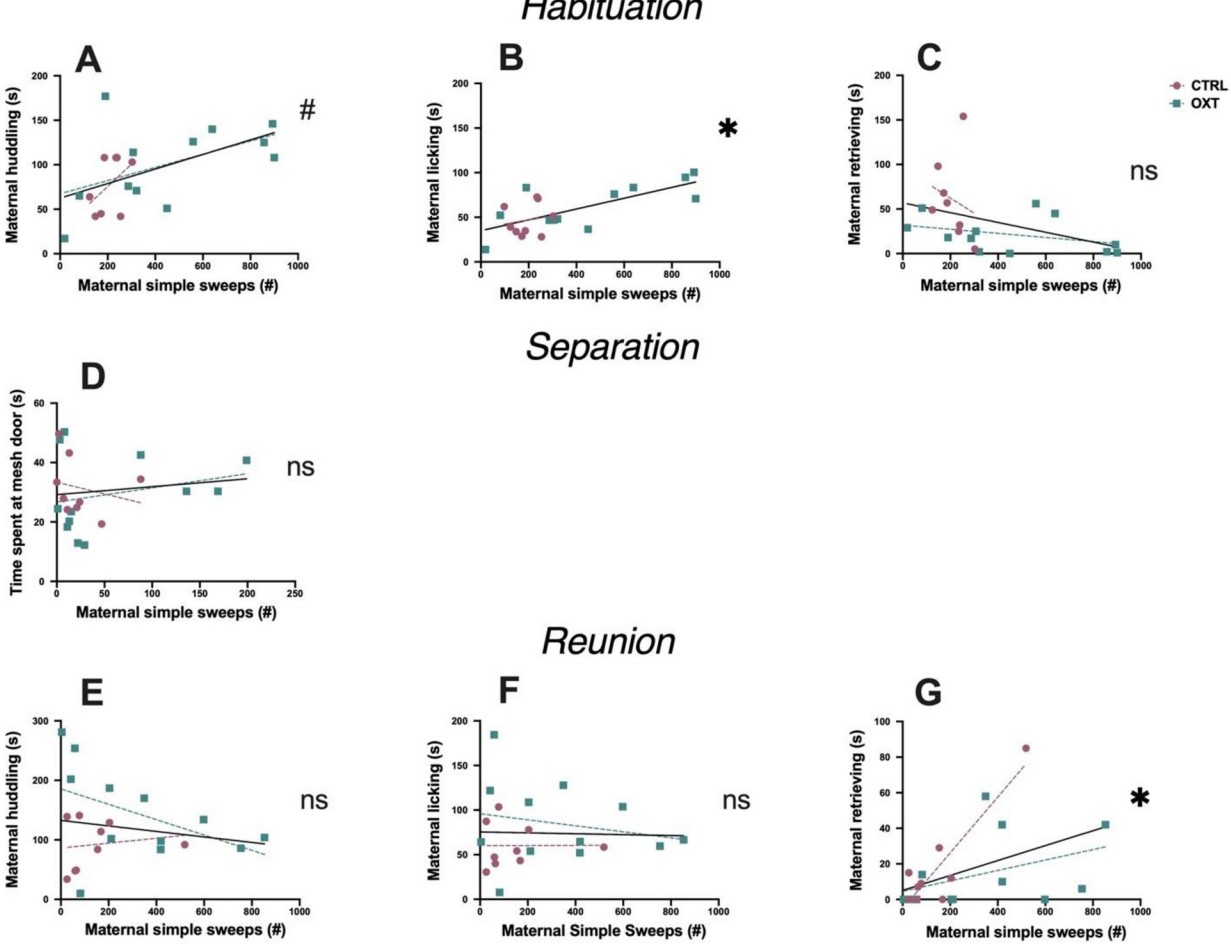

**Fig 4. Correlations between maternal simple sweep USVs and maternal care. (A-C) Habituation**. **(A)** There was a nonsignificant trend correlation between maternal simple sweeps and huddling. **(B)** Maternal simple sweeps positively correlated with licking behavior. **(C)** Maternal simple sweeps did not correlate with retrieving/carrying. Mothers given IN OXT carried/retrieved pups less than mothers given saline. **(D) Separation**. There was no correlation between time mothers spent at the mesh divider and maternal simple sweep USVs. **(E-G) Reunion**. **(E)** Maternal simple sweeps did not correlate with huddling or **(F)** licking. **(G)** Maternal simple sweeps positively correlated with maternal retrievals. Correlation line in black is the average slope across treatment groups; magenta and teal lines are the slopes for the saline and OXT treatments, respectively. *p<0.05; #p<0.10.

the mesh gate ($F_{1,20} = 0.89$, $p = 0.36$, $\Delta R^2 = 0.04$) (**Fig 5D**). Lastly, during the reunion phase, pup USVs positively correlated with retrieving/carrying ($F_{1,20} = 9.94$, $p = 0.016$, $\Delta R^2 = 0.34$) (**Fig 5G**) but was not correlated with either huddling ($F_{1,20} = 0.05$, $p = 0.823$, $\Delta R^2 = 0.003$) (**Fig 5E**) or licking ($F_{1,20} = 0.893$, $p = 0.36$, $\Delta R^2 = 0.043$) (**Fig 5F**). Across all correlations of USVs and maternal care, significant correlations occurred, but neither maternal simple sweep USVs nor pup whine USVs consistently correlated with a specific type of maternal care.

## Discussion

Maternal care and communication have lifelong consequences for offspring [83–86]. Therefore, it is important to elucidate the proximate hormonal mechanisms that increase maternal care and communication. OXT is known for its potent role in maternal physiology, neurophysiology, and social behavior, but whether OXT could rapidly change vocal production and behavior in mothers remained unknown. We aimed to fill these knowledge gaps by testing

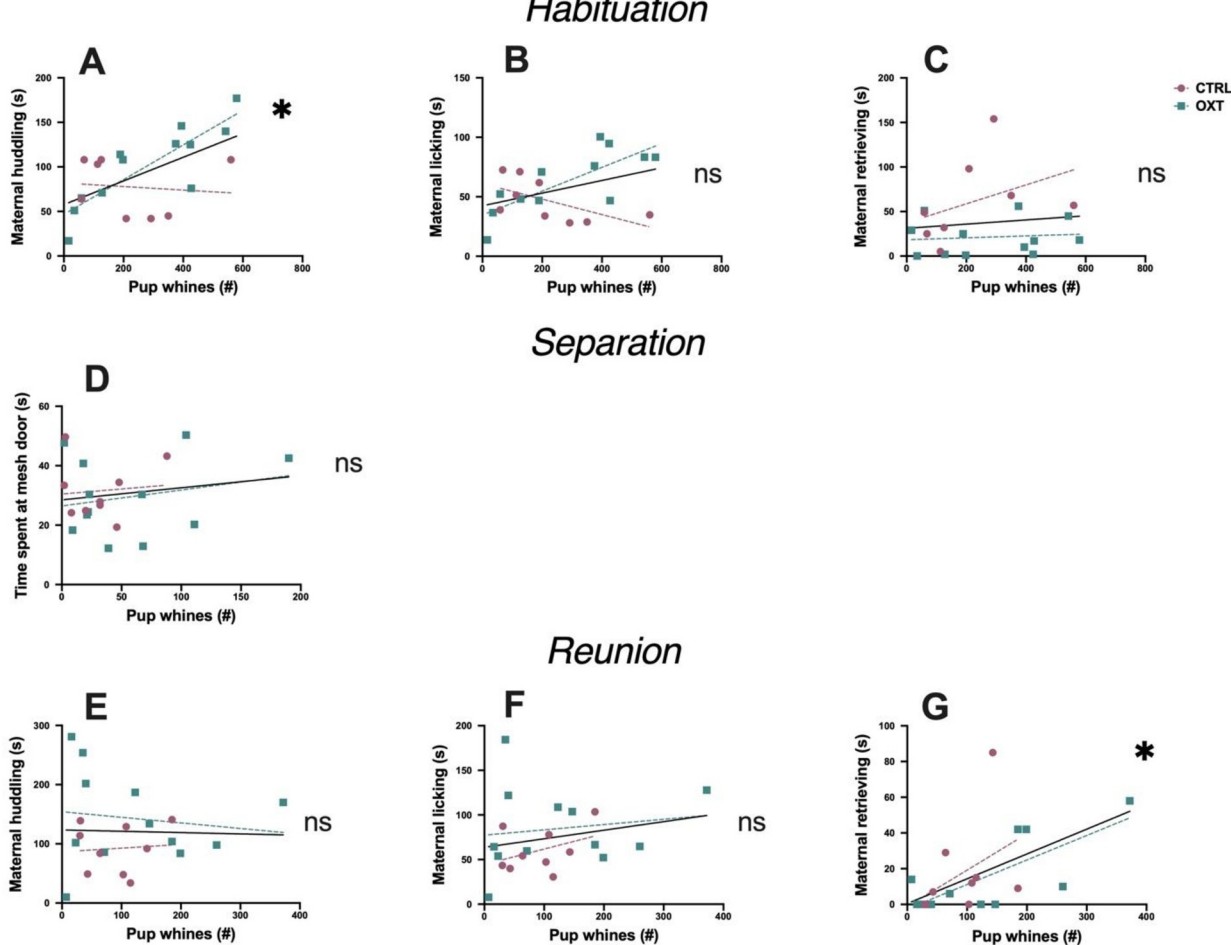

**Fig 5. Correlations between pup whine USVs and maternal care. (A-C) Habituation**. **(A)** There was a significant positive correlation between pup whines and maternal huddling. **(B)** There was no correlation between pup whines and maternal licking or **(C)** pup whines and maternal retrieval/carrying behavior. **(D) Separation**. Pup whines did not correlate with time that the mother spent at the mesh gate. **(E-G) Reunion**. **(E)** There was no correlation between pup whines and maternal huddling or **(F)** pup whines and maternal licking. **(G)** Pup USVs positively correlated with maternal retrieving/carrying. Correlation line in black is the average slope across treatment groups; magenta and teal lines are the slopes for the saline and OXT treatments, respectively. *p<0.05.

maternal response and effects of IN OXT during a potentially challenging and stressful pup separation paradigm.

We predicted that maternal care activity would be associated with vocal production and that IN OXT would amplify this effect. We found support for this hypothesis as all mothers produced more simple sweep USVs per second during both the habituation phase and reunion phase when mothers had access to tactile pup experience. We speculate that maternal sweeps decreased during the separation phase because mothers no longer had physical access to pups. The relative reduction of maternal simple sweeps during the separation phase may suggest that simple sweeps occur more frequently during social contact or may reflect a difference in internal state. In support of the social salience theory [87–91], IN OXT amplified the effect of the tactile pup experience, leading to an increase above and beyond the observed increase in control mothers during the habituation phase (and a trend in reunion phase). To our knowledge, this is the first study reporting context-dependent IN OXT-moderated changes in USV production that were associated with physical access to a social stimulus [92].

There are several possible functions for simple sweep USV production in the context of maternal care. Increased simple sweep USVs during maternal care may result from a higher state of arousal that is regulated by the autonomic nervous system [93]. This is supported by evidence in prairie voles, where vocal features covary with heart rate—longer vocalizations were associated with increased vagal tone and more calm behavior whereas shorter vocalizations were associated with decreased vagal tone and more anxious behavior [94]. Alternatively, increases in simple sweep USVs may be associated with a positive affective state [95]. In rats, 50 kHz USVs (similar kHz as California mouse simple sweeps) have been associated with positive affective state [9], and in California mice, simple sweeps typically occur during affiliative contexts [71]. This adds to a growing body of literature that aims to elucidate California mouse call type with function. Complex sweeps predict pair mate social behavior [46]; syllable USVs are associated with aggression (shorter calls) [44] and female preference (longer calls) [46]; both syllable USVs (shorter calls) and barks are associated with increased aggression [45]; pup whines can also elicit paternal retrievals [96]. As expected, we did not observe any syllable vocalizations or barks and only a handful of complex sweeps across all tests as these calls tend to be associated with aggression or adult interactions. Because IN OXT increased maternal simple sweep USVs, and in other studies, IN OXT has increased both vagal tone [97] and positive affective state [98–100], we suggest that maternal simple sweeps are most likely to be associated with changes in affective state.

In contrast to mothers, we did not expect to see a treatment effect with regards to pup vocalizations because only the mothers were treated. As expected, we did not observe differences in pup vocalizations between pup with control mothers and IN OXT mothers. This suggests that effects of IN OXT on the mother do not directly and rapidly influence pup behavior. Overall, pups called the most when first placed into the new chamber with their mothers at the rate of 0.94 whines per second, and then called at a steady rate of 0.33 whines per second during separation and reunion. This suggests that pup vocal production does not vary by social contact in the same way as maternal vocalizations. Instead, pup vocal production may be a function of their thermal challenge, as indicted by previous studies on rat pup USVs demonstrating pups increase USVs when first separated from their nest and given thermal challenges [101, 102]. After losing a certain amount of heat energy, number of pup whines produced may decrease to balance energy conservation and venous blood return to the heart.

We predicted a correlation between maternal simple sweeps and pup whines. We expected that the correlation between mother-pup USVs would be driven by the mother's vocalizations and/or behavior, as the pups' ear canals have not opened at PND 2–3, likely rendering them deaf [103]. Mother and pup USVs did not correlate during habituation, possibly because pup whine USVs were highest during this phase (**Fig 2B**) or possibly because the removal from the home cage at the start of the test disrupted the coordination between mothers and pups. There was also a nonsignificant trend for IN OXT to improve the correlation between mother and pup USVs. If the removal from the home cage at the start of the test disrupted the coordination between mothers and pups, IN OXT may be mediating this negative effect by increasing the salience of pup whine stimuli [104], allowing mothers to more effectively and efficiently cope with the challenge. During reunion, we found support for our initial prediction: maternal simple sweeps and pup whines positively correlated. Mothers may be more responsive to pup whines during the reunion phase because the pups have been without care for a longer period of time. An alternative explanation is that this synchrony occurred out of necessity because, with the second chamber open, the mice had double the space in the reunion phase compared to the habituation phase. In lambs and ewes, mother-offspring vocalizations have been shown to be important for recognition and location purposes [105, 106] with young lambs only being able to distinguish their mother via low frequency calls but using high frequency calls during

vocal exchanges [107]. Mother-offspring vocalizations that are contingent on each other's vocalizations have also been observed across cultures in humans [108]. To our knowledge, this is the first reporting of correlations between maternal USVs and offspring USVs in an animal model.

We also wanted to explore whether maternal simple sweeps or pup whines are correlated with specific types of maternal behavior. During habituation, USVs from both mothers and pups were associated with greater maternal care but were not associated with the same maternal behavior; maternal simple sweeps positively correlated with maternal licking behavior and pup whines positively correlated with maternal huddling behavior. During reunion, we saw a different relationship between USVs and maternal behavior, suggesting that if pup calls and not maternal responsiveness are driving these correlations, the pup whines are not eliciting a specific type of maternal care. During reunion, both maternal simple sweeps and pup whines positively correlated with maternal retrieval/carrying behavior. This suggests that after a separation event, pup whines may drive maternal retrieval/carrying behavior, similar to the finding previously reported in fathers [96], and that maternal simple sweeps may be a reliable signal supporting maternal responsiveness. Studies in the literature show support for this effect being driven by maternal responsiveness in mice [109, 110] and other studies show that the effect can also be driven by the pups [111, 112].

Finally, based on the prosocial effects of OXT, we predicted that an acute dose of IN OXT would increase maternal care. Our results show that in the context of our paradigm, IN OXT maintains maternal care rather than overtly increasing it. Mothers given IN OXT showed consistent number of retrievals pre- and post-separation while control mothers significantly decreased number of retrievals performed post-separation, leading to greater maintenance of total maternal care for IN OXT mothers. These findings are consistent with other studies in the literature in sheep [113], mice [114], rats [115, 116], and humans [117] but highlight that OXT can also increase maternal behavior within minutes of administration. We did not find that IN OXT led to a significant decrease in latency to approach pups after separation, but we found a non-significant trend. This supports previous findings in the literature that OXT has been associated with a reduction in the latency to retrieve or start maternal behavior [118, 119] though several other studies have not reported an effect of OXT on latency to retrieve pups [114, 120, 121]. Notably, we did not find any simple effects of IN OXT on nonmaternal behaviors during the habituation, separation, or reunion phases. This suggests that IN OXT specifically influences maternal behavior and not general activity (autogrooming, rearing) or anxiety (freezing) in female California mice. If IN OXT is dampening the stress/anxiety response, it is specific to maternal anxiety. This is important to note because one hypothesis regarding the effects of IN OXT is that it primarily functions as an anxiolytic agent versus a pro-social capacity across a variety of contexts and species [122–124]. In certain contexts, OXT can also have anxiogenic effects [62, 125]. However, we do not find that OXT is promoting anxiety in this context. Our results suggest that in this context, IN OXT has a specific effect on maternal care behavior that is not explained by differences in the non-maternal activities related to activity or stress.

In summary, these data are consistent with the concept that IN OXT rapidly and selectively increases maternal vocalizations and maintains maternal care. This data also highlights the importance of social contact for normal communication and care and enhancement by IN OXT. Overall, we propose that higher levels of OXT in mothers function to increase efficiency and maintain maternal care, particularly during challenges.

## Supporting information

**S1 Fig. Ethogram with description of behaviors measured.**
(TIF)

**S1 Table. Means and SEMs of behaviors measured.**
(TIF)

## Acknowledgments

We would like to thank undergraduate student support for their work during implementation of the experiment and quantification of behavior, and UW Madison animal care staff for their excellent care of the animals.

## Author Contributions

**Conceptualization:** Caleigh D. Guoynes.

**Data curation:** Caleigh D. Guoynes.

**Formal analysis:** Caleigh D. Guoynes.

**Funding acquisition:** Catherine A. Marler.

**Investigation:** Catherine A. Marler.

**Methodology:** Caleigh D. Guoynes.

**Project administration:** Caleigh D. Guoynes.

**Resources:** Catherine A. Marler.

**Supervision:** Catherine A. Marler.

**Writing – original draft:** Caleigh D. Guoynes.

**Writing – review & editing:** Caleigh D. Guoynes, Catherine A. Marler.

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
