## [Decision Letter · Decision Letter 0]

31 Dec 2020

PONE-D-20-37165

An acute dose of intranasal oxytocin rapidly increases maternal communication and maintains maternal care in primiparous postpartum California mice

PLOS ONE

Dear Dr. Guoynes,

Thank you for submitting your manuscript to PLOS ONE. After careful consideration, we feel that it has merit but does not fully meet PLOS ONE’s publication criteria as it currently stands. Therefore, we invite you to submit a revised version of the manuscript that addresses the points raised during the review process.

Thank you for submitting this intriguing study to Plos One. Three experts in the field have reviewed the manuscript, and all expressed considerable enthusiasm for the studies and conclusions. All three have provided helpful suggestions on how the work may be improved. Please consider whether other statistical models may be used, in particular using the litter as the statistical unit. Some more details in the introduction, as commented on by all reviewers, might be helpful. Please provide additional justification for only considering vocalizations in the ultrasonic range. The comment from Reviewer 2 about intranasal administration of oxytocin should be addressed as this route of administration has become more widely used.  

We look forward to receiving your revised manuscript.

Kind regards,

Cheryl S. Rosenfeld, DVM, PhD

Academic Editor

PLOS ONE

Journal Requirements:

2. To comply with PLOS ONE submissions requirements, in your Methods section, please provide additional information on the animal research and ensure you have included details on (1) methods of sacrifice, (2) methods of anesthesia and/or analgesia, and (3) efforts to alleviate suffering.

Reviewers' comments:

Reviewer's Responses to Questions

**Comments to the Author**

1. Is the manuscript technically sound, and do the data support the conclusions?

Reviewer #1: Yes

Reviewer #2: Yes

Reviewer #3: Yes

2. Has the statistical analysis been performed appropriately and rigorously? 

Reviewer #1: Yes

Reviewer #2: Yes

Reviewer #3: Yes

3. Have the authors made all data underlying the findings in their manuscript fully available?

Reviewer #1: Yes

Reviewer #2: No

Reviewer #3: Yes

4. Is the manuscript presented in an intelligible fashion and written in standard English?

Reviewer #1: Yes

Reviewer #2: Yes

Reviewer #3: Yes

5. Review Comments to the Author

Reviewer #1: In this manuscript the authors describe the results of behavioral experiments examining the effects of intranasal oxytocin on maternal behavior and vocalizations in female California mice. Unlike other rodent species, in which the function of vocalizations are not well described, vocalizations for California mice have been well characterized in the lab and in the field. Furthermore this is one of the few papers to examine how vocalizations are used by a mother to interact with pups. The authors find that intranasal oxytocin enhance maternal vocalizations in the presence of pups but not when separated from pups. This suggests that oxytocin may enhance salience of pup interactions for a mother. Overall the manuscript is well organized. The methods were detailed and experimental procedures were justified. However there are several points that need clarified and some adjustments to statistical analyses might give the authors more power to detect differences. Overall this is a very strong contribution.

Introduction

The authors state that preliminary data indicate that maternal simple sweeps are primarily used in interactions with pups. This is a pretty definitive statement at this point in the manuscript and I was expecting more information about what these vocalizations are and why the authors think they are not used in other contexts. I suggest the authors consider moving some of the details of these call described in the methods to the introduction.

Did the authors predict that oxytocin would affect behaviors during specific stages of the behavioral testing.

Methods

On line 111 the authors should revise “neuroendocrine responses” to “glucocorticoid responses”, as regulation of oxytocin and corticosterone have distinct mechanisms of regulation.

On line 136, reasonable people could disagree that the intranasal administration, which involves scruffing the mouse, is less stressful than an IP or SC injection.

The authors could also state that the behavioral effects of intranasal oxytocin in California mice are consistent with the outcomes of central oxytocin manipulations (Duque-Wilckens et al. 2018, 2020), which also suggests that intranasal OT accesses the brain.

Line 207-208: revise to observer blind to treatment

Line 264-265: the authors say the total number of vocalizations was recorded and used “for comparison”. Please me more specific. Were variables of interest normalized to this variable or was it analyzed separately? Did the total number of vocalizations differ between the two groups?

The authors report several statistical tests that are at nonsignificant trend level. The authors might want to consider correcting for the False Discovery Rate instead of using Bonferroni correction since this approach protects against type I error without sacrificing as much statistical power.

Results

It could be useful to include the age of the pups when intranasal oxytocin/behavioral observations are performed in Figure 1. It might also be more intuitive for the authors to label the microphones for reader.

Please increase the fonts of the axes and labels on all bar and scatter plots, they are hard to read.

Figure 2A: significance symbols are not intuitive, even after checking the figure legend. Specifically, the * comparing the different stages of the test is confusing. The labeling looks like habituation and reunion are different but I think the authors are trying to show that separation is different from the other groups. Consider replacing with a different symbol like a dagger and additional notation to indicate which groups are different.

For figure 3: I suggest the authors discuss behaviors with significant differences first (perhaps total maternal care and then individual behaviors.

On line 355 the authors state that high levels of retrieval are associated with less efficient maternal care. Are studies cited conducted in the home cage or in a novel environment? Effects of oxytocin are context dependent. In virgin female California mice intranasal oxytocin is anxiogenic in a novel environment but anxiolytic in the home cage. The authors should address the role of context when interpreting these results and also the extent to which effects of oxytocin change in dams versus virgins.

For scatterplots, separate regression lines for control and oxytocin groups should be plotted. In most cases these lines will overlap but they are important for the few variables where these slopes differ (eg Fig. 4C).

Discussion

Lines 452-453: the authors should be more precise when describing their results, as only a few key variables were affected by intranasal oxytocin.

Some parts of the discussion are repetitive (pups being deaf is mentioned 2x). Also, why do the authors think that sweeps are decreased in the separation phase.

Line 530: the authors should be more precise in their language. There was no increase in maternal care in the oxytocin group, only the absence of a decrease.

Line 538-547: When discussing effects of oxytocin on anxiety the authors should be aware that oxytocin can have anxiogenic effects, particularly in females/women.

Last line of discussion: This sentence is not in line with the conclusions the authors make earlier in the manuscript where they correctly observe that intranasal oxytocin maintains maternal behavior but does increase it relative to the habituation stage.

Reviewer #2: This article examines the effects of intranasal oxytocin on maternal behaviors, including maternal vocalizations (as well as pup vocalizations), in California mice. It is a well-done study and should be of broad interest to those interested in the neurobiology of maternal behavior – including those interested in human maternal behavior. I do have a few comments below intended to help improve the paper.

Introduction:

Lines 46-47: This sentence is an awkward introduction to neural substrates – they don’t really “complement” vocalizations as they are also involved in generating vocalizations.

Line 81: Not sure that there is actual evidence that these genotypes affect oxytocin receptor availability. I think that it is a presumed relationship.

Line 89: California mice are a great model but not sure why they are “uniquely” suited. Since this is a study of maternal behavior, seems like many rodent species would be suitable.

Lines 96-103: Perhaps predicted effects on pup vocalizations should also be mentioned here (later it is said that no effects were predicted, but that’s not clear here).

Methods:

Line 137: Is this supposed to be “CSF AND plasma concentrations of OXT”?

How many pups were transferred with each mother? All of them? What was the average number of pups for females that got IN OXT vs females which got saline?

Line 207: I would change this to “an observer blind to treatment condition”.

Lines 250-261: The sentence regarding Bonferroni corrections is in this paragraph twice, one can be removed.

Figure 2 legend, section b. “Pups” at the beginning of sentence 3 is not capitalized.

Results:

I think that it is excellent that the data were recorded in such a way that the temporal sequence of mother and pup calls could be synced. Unfortunately, the current analysis does not seem to take full advantage of these data. A time-series analysis, or some type of dyadic analysis, would allow the authors to dig deeper into the interactions between mothers and pups, and the effects of oxytocin on those analyses.

Discussion:

Lines 477-488. I’m not sure that I follow the reasoning here, at least regarding thermal challenge. Wouldn’t thermal conditions alter between separation and reunion, suggesting that USVs should alter as well?

Lines 490-491. Don’t need to repeat the part about pups probably being deaf (it’s also in lines 477-479). Also, even if pups can’t hear, is it possible that the mothers’ bodies would vibrate differently depending on their types of vocalizations? This presumably could be detected by the pups.

Lines 539-547. Please be specific here about what behaviors you are designating as potential “anxiety” behaviors during this test.

Lines 548-549. I thought you just said above that IN OXT maintains rather than increases maternal behavior…

Data Availability

The authors say that all the data are available within the manuscript or supporting information…I don’t see raw data, just means and standard errors.

Reviewer #3: This manuscript describes a very interesting and innovative study investigating the effects of intranasal oxytocin on mothers’ behavioral and vocal responses to their pups, as well as on the pups’ vocalizations, in the California mouse. The study was well designed, and the manuscript is clearly written, for the most part.

My main comments and suggestions are as follows:

Given that California mouse adults and pups produce calls in the audible spectrum, in addition to USVs, it would be helpful to include a rationale for focusing exclusively on USVs.

I was surprised that number of pups per litter was not used as a covariate or, as far as I could tell, used in any other way in analyses. Isn’t this likely to be a major determinant of the number of pup vocalizations produced as well as, perhaps, the amount of maternal care performed?

In the introduction, it would be helpful to provide some information on the time course by which IN OXT has been found to influence behavior in other studies. The specific time course in this study is mentioned briefly in the discussion (L533-534) but nowhere else.

Minor comments:

Figs. 2-5 – I found it very difficult to read the text on the y-axes. Consider putting at least part of this info (e.g., the name of the behavior) above each graph.

L46-47 – The statement that “neural substrates also play an important role…” is so obvious as to be unnecessary, since virtually all mammalian behaviors require neural substrates.

L51-52 – This sentence is somewhat misleading. As written, it suggests that the increase in plasma OXT was co-opted evolutionarily for facilitating maternal care. However, it was presumably the increase in central OXT, rather than in plasma OXT per se, that was co-opted.

L54 – It would be more appropriate to say that estrogen levels prime the neural substrates that respond to OXT, rather than that they prime OXT itself.

L61 – Add “females” after “low grooming” (and possibly after “high grooming”).

L63 – Can you be more specific about the reproductive state of non-lactating females – e.g., were they virgins?

L66 – Change “connect” to “connects” and “activate” to “activates” (if these words refer to dopamine neurons rather than to the AVPV).

L69 – Should “efficiency” be changed to “efficacy”? Either way, I’m not convinced that this statement is supported by the findings summarized above.

L71 – “their” can be deleted.

L89 – Please add the Latin name (currently it’s given a little later in the manuscript).

L89 – The phrase “uniquely suited” seems too strong; certainly, many other species have similarly complex vocal repertoires and could therefore be excellent model species for this sort of work.

For me, it would be helpful to be consistent in referring to the pup calls that were monitored as either “pup USVs” or “pup whines” rather than using both phrases. I realize, though, that other readers might not find this lack of consistency to be a problem.

L94-95 – The statement that “maternal simple sweeps and pup wines are the primary maternal USVs” is confusing, as it seems to suggest that pup whines are produced by mothers, not by pups. Remove “maternal” before “USVs”?

L101-103 – To use parallel sentence structure, either remove “enhance” in L102 or add a verb before “maternal USV production”.

L110 – Delete “stress” – dexamethasone isn’t a stressor! “Stress” can be replaced with “challenge,” but this isn’t necessary.

L110 – Change “between” to “in” or “within”.

L108-111 – It’s not clear how this statement is relevant to this paragraph. Yes, females across this age range have been shown to have similar cort responses to CRH and DEX, but that’s not necessarily applicable to “neuroendocrine responses” in general.

L114 – Confusing what they were randomly assigned to. Do you mean that female and male pairmates were randomly assigned to one another or that the pairs were randomly assigned to something (presumably experimental condition)?

L116 – Remove “Female”.

L120 – Change “Treatments” to something like “Mothers” or “Mice” or “Subjects”.

L129-130 – Was the cannula needle alternated between nostrils within individual animals or across animals?

L130 – “Mice” in this line presumably refers to house mice (Mus) and not California mice. Please clarify this, as some readers won’t know that California mice aren’t “real mice” phylogenetically.

L138 – Change “human” to “humans”.

L143 – Delete “that”.

L155 – Harris et al. 2017 isn’t in the reference list, and it’s not clear what paper is being referred to.

L156-157 – The relevance of this sentence about blunted behavioral responses in parents isn’t clear.

L260-261 – This sentence is identical to one earlier in the paragraph. Delete.

L266-267 – I don’t understand why t-tests were used to examine main effects of treatment. Wasn’t the effect of treatment discernible from the 2-way ANOVAs, which I assume examined the effects of test phase and treatment?

L346-347 – This sentence is confusing for two reasons. First, it’s not clear which test phase it refers to (I assume it’s the beginning of the reunion phase, but that should be stated explicitly). Second, the fact that it comes right after two sentences about the use of delta scores seems to suggest that this sentence, too, refers to delta scores, but that’s not the case.

L355-357 – Please indicate which species this sentence refers to.

L385 – The subheading sounds awkward. Consider changing to “Correlations of maternal care with maternal USVs and pup USVs”.

L424-425 – The phrasing “had no effect on…” strikes me as odd, since this section talks about correlations rather than causal relationships. Change to “was not correlated with”?

L438 – Omit “later in life” since this is implicit in “lifelong”?

L452-453 – Not clear what “the control increase in the habituation phase” refers to.

L455 – Marler & Monari 2020 is not in the reference list.

L485-488 – Please specify the species being discussed.

L490 – The phrase “driven by the mother” isn’t clear, because it could refer to either the mother’s vocalizations on the mother’s responses to the pups’ vocalizations. Please clarify.

L504 – The phrase “sheep and ewes” is redundant. Say either “lambs and ewes” or simply “sheep”.

L506 – Change “ewes” to “lambs”.

L507 – I don’t understand the phrase “but using high frequency calls to during vocal exchanges.” Is “to” meant to be “too” or is there a missing word?

L521 – Add a comma after the reference.

L522-525 – Please specify what species are being referred to.

L550-552 – If I understand this sentence correctly, it’s meant to refer to higher endogenous levels of OXT in mothers than in other females. If so, then “IN” should be removed.

Fig. 3D caption – Indicate that this treatment effect was not seen in Controls, or that the net positive effect of IN OXT was significant relative to Controls?

Supp. Fig. 1 – The first three definitions contain typos.

Supp. Fig. 1 – It would seem that mothers often meet the criteria for “freezing” while huddling their pups. Are these two behaviors mutually exclusive? Please clarify.

6. PLOS authors have the option to publish the peer review history of their article (what does this mean?). If published, this will include your full peer review and any attached files.

Reviewer #1: No

Reviewer #2: No

Reviewer #3: No

---

## [Author Response · Author response to Decision Letter 0]

23 Feb 2021

Reviewer #1: In this manuscript the authors describe the results of behavioral experiments examining the effects of intranasal oxytocin on maternal behavior and vocalizations in female California mice. Unlike other rodent species, in which the function of vocalizations are not well described, vocalizations for California mice have been well characterized in the lab and in the field. Furthermore this is one of the few papers to examine how vocalizations are used by a mother to interact with pups. The authors find that intranasal oxytocin enhance maternal vocalizations in the presence of pups but not when separated from pups. This suggests that oxytocin may enhance salience of pup interactions for a mother. Overall the manuscript is well organized. The methods were detailed and experimental procedures were justified. However there are several points that need clarified and some adjustments to statistical analyses might give the authors more power to detect differences. Overall this is a very strong contribution.

Introduction

The authors state that preliminary data indicate that maternal simple sweeps are primarily used in interactions with pups. This is a pretty definitive statement at this point in the manuscript and I was expecting more information about what these vocalizations are and why the authors think they are not used in other contexts. I suggest the authors consider moving some of the details of these call described in the methods to the introduction.

-L102-109. Clarified the statement to include that previous recordings suggested maternal simple sweeps and pup whines were the primary call types used in the context of mother-pup interactions (vs. complex sweeps, SVs, barks), and that both simple sweeps and pup whines can occur in other social contexts

Did the authors predict that oxytocin would affect behaviors during specific stages of the behavioral testing.

-L121-122. Clarified to predict that we predicted separation of mother and pups would disrupt pro-social effects of OXT

Methods

On line 111 the authors should revise “neuroendocrine responses” to “glucocorticoid responses”, as regulation of oxytocin and corticosterone have distinct mechanisms of regulation.

-L135. Amended 

On line 136, reasonable people could disagree that the intranasal administration, which involves scruffing the mouse, is less stressful than an IP or SC injection.

-L171. Removed IP injection

The authors could also state that the behavioral effects of intranasal oxytocin in California mice are consistent with the outcomes of central oxytocin manipulations (Duque-Wilckens et al. 2018, 2020), which also suggests that intranasal OT accesses the brain.

-177-179. Amended

Line 207-208: revise to observer blind to treatment

-L253. Amended

Line 264-265: the authors say the total number of vocalizations was recorded and used “for comparison”. Please me more specific. Were variables of interest normalized to this variable or was it analyzed separately? Did the total number of vocalizations differ between the two groups?

-L313-314. Amended to reflect USVs/sec was used to account for time differences in the contact vs separation phases ( 5 min vs. 3 min)

The authors report several statistical tests that are at nonsignificant trend level. The authors might want to consider correcting for the False Discovery Rate instead of using Bonferroni correction since this approach protects against type I error without sacrificing as much statistical power.

-L301-302. Amended corrections to use the Benjamini-Hochberg False Discovery Rate correction, but unfortunately the trending results were still trending (albeit closer to significant).

Results

It could be useful to include the age of the pups when intranasal oxytocin/behavioral observations are performed in Figure 1. It might also be more intuitive for the authors to label the microphones for reader.

-Amended

Please increase the fonts of the axes and labels on all bar and scatter plots, they are hard to read.

-Amended to increase axes size

Figure 2A: significance symbols are not intuitive, even after checking the figure legend. Specifically, the * comparing the different stages of the test is confusing. The labeling looks like habituation and reunion are different but I think the authors are trying to show that separation is different from the other groups. Consider replacing with a different symbol like a dagger and additional notation to indicate which groups are different.

-L371-372. Amended

For figure 3: I suggest the authors discuss behaviors with significant differences first (perhaps total maternal care and then individual behaviors.

-We did not make this change because we were concerned that if we make total maternal care the first graph, that readers may not realize that most of the difference in maternal care is coming from the change in retrievals. We would be fine changing this if the reviewer insists. 

On line 355 the authors state that high levels of retrieval are associated with less efficient maternal care. Are studies cited conducted in the home cage or in a novel environment? Effects of oxytocin are context dependent. In virgin female California mice intranasal oxytocin is anxiogenic in a novel environment but anxiolytic in the home cage. The authors should address the role of context when interpreting these results and also the extent to which effects of oxytocin change in dams versus virgins.

-L414-416. Amended—these studies do not specifically manipulate the OXT system, they just examine amount of time or number of retrievals on a desired maternal outcome (all pups in nest, pups located and taken back to nest)

For scatterplots, separate regression lines for control and oxytocin groups should be plotted. In most cases these lines will overlap but they are important for the few variables where these slopes differ (eg Fig. 4C).

-Added dashed line for each treatment

Discussion

Lines 452-453: the authors should be more precise when describing their results, as only a few key variables were affected by intranasal oxytocin.

-L523. Amended

Some parts of the discussion are repetitive (pups being deaf is mentioned 2x). Also, why do the authors think that sweeps are decreased in the separation phase.

-L513-520. Added speculation on why fewer simple sweeps occur during separation

Line 530: the authors should be more precise in their language. There was no increase in maternal care in the oxytocin group, only the absence of a decrease.

-L611. Amended

Line 538-547: When discussing effects of oxytocin on anxiety the authors should be aware that oxytocin can have anxiogenic effects, particularly in females/women.

-L630-632. Amended to add references for anxiogenic effects of OXT

Last line of discussion: This sentence is not in line with the conclusions the authors make earlier in the manuscript where they correctly observe that intranasal oxytocin maintains maternal behavior but does increase it relative to the habituation stage.

- L636. Amended

Reviewer #2: This article examines the effects of intranasal oxytocin on maternal behaviors, including maternal vocalizations (as well as pup vocalizations), in California mice. It is a well-done study and should be of broad interest to those interested in the neurobiology of maternal behavior – including those interested in human maternal behavior. I do have a few comments below intended to help improve the paper.

Introduction:

Lines 46-47: This sentence is an awkward introduction to neural substrates – they don’t really “complement” vocalizations as they are also involved in generating vocalizations.

-L46-47. Amended

Line 81: Not sure that there is actual evidence that these genotypes affect oxytocin receptor availability. I think that it is a presumed relationship.

-L89. Amended

Line 89: California mice are a great model but not sure why they are “uniquely” suited. Since this is a study of maternal behavior, seems like many rodent species would be suitable.

-L98. Amended

Lines 96-103: Perhaps predicted effects on pup vocalizations should also be mentioned here (later it is said that no effects were predicted, but that’s not clear here).

-L116-117. Amended

Methods:

Line 137: Is this supposed to be “CSF AND plasma concentrations of OXT”?

-L172. Amended

How many pups were transferred with each mother? All of them? What was the average number of pups for females that got IN OXT vs females which got saline?

-L141-145. All pups were added—amended methods to include mean and discussion of potential use in models as a covariate. In this experiment, adding pup number as a covariate did not explain additional variance in the models and reduced statistical power.

Line 207: I would change this to “an observer blind to treatment condition”.

-L253. Amended

Lines 250-261: The sentence regarding Bonferroni corrections is in this paragraph twice, one can be removed.

-L309. Amended

Figure 2 legend, section b. “Pups” at the beginning of sentence 3 is not capitalized.

-L361. Amended

Results:

I think that it is excellent that the data were recorded in such a way that the temporal sequence of mother and pup calls could be synced. Unfortunately, the current analysis does not seem to take full advantage of these data. A time-series analysis, or some type of dyadic analysis, would allow the authors to dig deeper into the interactions between mothers and pups, and the effects of oxytocin on those analyses.

When scoring behavior, we had another measure record: “contact” or “no contact” throughout the habituation and reunion phases, but this data was because the behavioral videos could only reasonably be scored by hand to the second, but in one second, there may be a bout of up to 20 sweep calls and it was difficult to decide whether to code this into the “contact” or “no contact” if it occurred during a behavior transition. Cutting out the time around a behavioral transition was considered, but this left more missing data for animals that changed their behavior frequently (from retrieving to rearing to huddling to walking) with less data than those mice that spent the majority of their time engaged in mostly maternal behavior (“contact”). We agree that it would fascinating to get a better look at the temporal sequence of calls and behavior, but I think that this would be best paired with automatic software behavioral coding to get an even more precise temporal sequence.

Discussion:

Lines 477-488. I’m not sure that I follow the reasoning here, at least regarding thermal challenge. Wouldn’t thermal conditions alter between separation and reunion, suggesting that USVs should alter as well?

-L558-563. Amended to add that calls in response to thermal challenges happen at first (i.e. first few minutes). 

Lines 490-491. Don’t need to repeat the part about pups probably being deaf (it’s also in lines 477-479). Also, even if pups can’t hear, is it possible that the mothers’ bodies would vibrate differently depending on their types of vocalizations? This presumably could be detected by the pups.

-L552. Removed the second ref to pups being deaf. We also wondered about pups ability to detect vibrations, but could not find sources to validate this assumption

Lines 539-547. Please be specific here about what behaviors you are designating as potential “anxiety” behaviors during this test.

-L626-627. Amended

Lines 548-549. I thought you just said above that IN OXT maintains rather than increases maternal behavior…

-L636. Amended

Data Availability

The authors say that all the data are available within the manuscript or supporting information…I don’t see raw data, just means and standard errors.

-Uploading data to Open Science Framework

Reviewer #3: This manuscript describes a very interesting and innovative study investigating the effects of intranasal oxytocin on mothers’ behavioral and vocal responses to their pups, as well as on the pups’ vocalizations, in the California mouse. The study was well designed, and the manuscript is clearly written, for the most part.

My main comments and suggestions are as follows:

Given that California mouse adults and pups produce calls in the audible spectrum, in addition to USVs, it would be helpful to include a rationale for focusing exclusively on USVs.

L282-283. Added a section to the methods section to include that some harmonics in the pup whine USVs can be heard in the audible spectrum. There is a note earlier in the methods section explains the only calls that we saw from mothers during this test were the simple sweep calls. Mothers can make calls in the audible frequency range (the SV calls), but we did not observe these call types in the audio files.

I was surprised that number of pups per litter was not used as a covariate or, as far as I could tell, used in any other way in analyses. Isn’t this likely to be a major determinant of the number of pup vocalizations produced as well as, perhaps, the amount of maternal care performed?

L141-150. Excellent suggestion. I added pup number as a covariate to the models, but it did not explain any additional variance and using the extra parameter cost statistical power in several of the models. Below is the additional information I included in the text:

 Number of pups was considered for use as a covariate, but in the statistical models, including this variable a) did not explain additional variance and b) reduced the power of the statistical comparison. Pup number across treatments was very similar—average number of pups for mothers in the saline condition was 2.11, and average number of pups for mothers in the OXT condition was 1.91.

In the introduction, it would be helpful to provide some information on the time course by which IN OXT has been found to influence behavior in other studies. The specific time course in this study is mentioned briefly in the discussion (L533-534) but nowhere else.

-L113-117. Added time course information to intro

Minor comments:

Figs. 2-5 – I found it very difficult to read the text on the y-axes. Consider putting at least part of this info (e.g., the name of the behavior) above each graph.

-Amended axes to be larger

L46-47 – The statement that “neural substrates also play an important role…” is so obvious as to be unnecessary, since virtually all mammalian behaviors require neural substrates.

-L46-47. Amended to combine two sentences and remove neural substrates

L51-52 – This sentence is somewhat misleading. As written, it suggests that the increase in plasma OXT was co-opted evolutionarily for facilitating maternal care. However, it was presumably the increase in central OXT, rather than in plasma OXT per se, that was co-opted.

-L53. Removed plasma

L54 – It would be more appropriate to say that estrogen levels prime the neural substrates that respond to OXT, rather than that they prime OXT itself.

-L57. Amended

L61 – Add “females” after “low grooming” (and possibly after “high grooming”).

-L64. Amended

L63 – Can you be more specific about the reproductive state of non-lactating females – e.g., were they virgins?

-L66. Amended—they were not virgins and were both maternally-experienced

L66 – Change “connect” to “connects” and “activate” to “activates” (if these words refer to dopamine neurons rather than to the AVPV).

-L69. Amended

L69 – Should “efficiency” be changed to “efficacy”? Either way, I’m not convinced that this statement is supported by the findings summarized above.

-L72. Amended wording to coordinating

L71 – “their” can be deleted.

-L74. Amended

L89 – Please add the Latin name (currently it’s given a little later in the manuscript).

-L-97. Amended

L89 – The phrase “uniquely suited” seems too strong; certainly, many other species have similarly complex vocal repertoires and could therefore be excellent model species for this sort of work.

-L-98. Amended to “well-suited”

For me, it would be helpful to be consistent in referring to the pup calls that were monitored as either “pup USVs” or “pup whines” rather than using both phrases. I realize, though, that other readers might not find this lack of consistency to be a problem.

-Pup USVs across species can vary in structure significantly. For P. cal, pup USVs, we typically refer to them as “pup whines” whereas other species like rats and mice refer to the calls as “pup USVs”

L94-95 – The statement that “maternal simple sweeps and pup wines are the primary maternal USVs” is confusing, as it seems to suggest that pup whines are produced by mothers, not by pups. Remove “maternal” before “USVs”?

- L102-109. Amended to clarify, add more context and references

L101-103 – To use parallel sentence structure, either remove “enhance” in L102 or add a verb before “maternal USV production”.

-L120. Amended

L110 – Delete “stress” – dexamethasone isn’t a stressor! “Stress” can be replaced with “challenge,” but this isn’t necessary.

-L129. Amended

L110 – Change “between” to “in” or “within”.

-L129. Amended

L108-111 – It’s not clear how this statement is relevant to this paragraph. Yes, females across this age range have been shown to have similar cort responses to CRH and DEX, but that’s not necessarily applicable to “neuroendocrine responses” in general.

-L-135. Amended to glucocorticoid

L114 – Confusing what they were randomly assigned to. Do you mean that female and male pairmates were randomly assigned to one another or that the pairs were randomly assigned to something (presumably experimental condition)?

-L-138. Amended to reflect that pairings were random

L116 – Remove “Female”.

-L140. Amended

L120 – Change “Treatments” to something like “Mothers” or “Mice” or “Subjects”.

-L148. Amended

L129-130 – Was the cannula needle alternated between nostrils within individual animals or across animals?

-L164. Clarified that it was within individuals

L130 – “Mice” in this line presumably refers to house mice (Mus) and not California mice. Please clarify this, as some readers won’t know that California mice aren’t “real mice” phylogenetically.

-L164. Amended to rodents

L138 – Change “human” to “humans”.

-Amended

L143 – Delete “that”.

-L180. Deleted

L155 – Harris et al. 2017 isn’t in the reference list, and it’s not clear what paper is being referred to.

-L196. Added Harris BN, Perea-Rodriguez JP, Saltzman W. Acute effects of corticosterone injection on paternal behavior in California mouse (Peromyscus californicus) fathers. Hormones and Behavior. 2011 Nov 1;60(5):666-75.

L156-157 – The relevance of this sentence about blunted behavioral responses in parents isn’t clear.

-L201. This is just more evidence that typical stressors and novel scents (even if potentially warning of danger) do not elicit a strong stress response

L260-261 – This sentence is identical to one earlier in the paragraph. Delete.

-L309. Deleted.

L266-267 – I don’t understand why t-tests were used to examine main effects of treatment. Wasn’t the effect of treatment discernible from the 2-way ANOVAs, which I assume examined the effects of test phase and treatment?

-L334. The ANOVA results and t-test results were very similar, but we decided to use the reunion-habituation change score and used t-tests for those findings (Figure 3 vs. Figure 2).

L346-347 – This sentence is confusing for two reasons. First, it’s not clear which test phase it refers to (I assume it’s the beginning of the reunion phase, but that should be stated explicitly). Second, the fact that it comes right after two sentences about the use of delta scores seems to suggest that this sentence, too, refers to delta scores, but that’s not the case.

-L403. Amended

L355-357 – Please indicate which species this sentence refers to.

-L414. Amended

L385 – The subheading sounds awkward. Consider changing to “Correlations of maternal care with maternal USVs and pup USVs”.

-L446. Amended

L424-425 – The phrasing “had no effect on…” strikes me as odd, since this section talks about correlations rather than causal relationships. Change to “was not correlated with”?

-L489. Amended

L438 – Omit “later in life” since this is implicit in “lifelong”?

-L502. Amended

L452-453 – Not clear what “the control increase in the habituation phase” refers to.

-L523. Amended

L455 – Marler & Monari 2020 is not in the reference list.

-L526. Amended

L485-488 – Please specify the species being discussed.

-L559. Amended

L490 – The phrase “driven by the mother” isn’t clear, because it could refer to either the mother’s vocalizations on the mother’s responses to the pups’ vocalizations. Please clarify.

-L565-569. Amended

L504 – The phrase “sheep and ewes” is redundant. Say either “lambs and ewes” or simply “sheep”.

-L582. Amended

L506 – Change “ewes” to “lambs”.

-L584. Amended

L507 – I don’t understand the phrase “but using high frequency calls to during vocal exchanges.” Is “to” meant to be “too” or is there a missing word?

-L585. Removed “to”

L521 – Add a comma after the reference.

-L602. Amended

L522-525 – Please specify what species are being referred to.

-L604. Amended

L550-552 – If I understand this sentence correctly, it’s meant to refer to higher endogenous levels of OXT in mothers than in other females. If so, then “IN” should be removed.

-L638. Amended

Fig. 3D caption – Indicate that this treatment effect was not seen in Controls, or that the net positive effect of IN OXT was significant relative to Controls?

-Amended

Supp. Fig. 1 – The first three definitions contain typos.

-Amended

Supp. Fig. 1 – It would seem that mothers often meet the criteria for “freezing” while huddling their pups. Are these two behaviors mutually exclusive? Please clarify.

-Amended

---

## [Decision Letter · Decision Letter 1]

19 Mar 2021

PONE-D-20-37165R1

An acute dose of intranasal oxytocin rapidly increases maternal communication and maintains maternal care in primiparous postpartum California mice

PLOS ONE

Dear Dr. Guoynes,

Thank you for submitting your manuscript to PLOS ONE. After careful consideration, we feel that it has merit but does not fully meet PLOS ONE’s publication criteria as it currently stands. Therefore, we invite you to submit a revised version of the manuscript that addresses the points raised during the review process.

Thank you for submitting this revised paper. The three original reviewers were very positive about the updated work. There are a few minor corrections and clarifications suggested before it can be considered for publication. Appreciate if you could respond to these suggestions. A final decision can then be made.

We look forward to receiving your revised manuscript.

Kind regards,

Cheryl S. Rosenfeld, DVM, PhD

Academic Editor

PLOS ONE

Journal Requirements:

Reviewers' comments:

Reviewer's Responses to Questions

**Comments to the Author**

1. If the authors have adequately addressed your comments raised in a previous round of review and you feel that this manuscript is now acceptable for publication, you may indicate that here to bypass the “Comments to the Author” section, enter your conflict of interest statement in the “Confidential to Editor” section, and submit your "Accept" recommendation.

Reviewer #1: All comments have been addressed

Reviewer #2: All comments have been addressed

Reviewer #3: All comments have been addressed

2. Is the manuscript technically sound, and do the data support the conclusions?

Reviewer #1: Yes

Reviewer #2: Yes

Reviewer #3: Yes

3. Has the statistical analysis been performed appropriately and rigorously? 

Reviewer #1: Yes

Reviewer #2: Yes

Reviewer #3: Yes

4. Have the authors made all data underlying the findings in their manuscript fully available?

Reviewer #1: Yes

Reviewer #2: Yes

Reviewer #3: Yes

5. Is the manuscript presented in an intelligible fashion and written in standard English?

Reviewer #1: Yes

Reviewer #2: Yes

Reviewer #3: Yes

6. Review Comments to the Author

Reviewer #1: All of my comments have been addressed in the revisions. The authors did a very nice job, this is a great contribution.

Reviewer #2: (No Response)

Reviewer #3: The authors were responsive to my previous comments, and the manuscript has improved. However, I have quite a few additional suggestions for increasing the clarify of the writing.

L12 – Mention saline controls.

L19 - Not clear what “Maternal-pup USVs were correlated upon reunion” means – the number of USVs? Frequency? Timing? (Also, “Maternal-pup USVs” itself isn’t clear; try using something like “Mothers’ and pups’ USVs” instead.)

L20 – “more positive change” is confusing. Change to something like “larger increase”?

L24 – Change “single dose of” to “single treatment with”?

L32 – Explain “pup whines.”

L46-48 – It’s somewhat misleading to say that OXT “likely plays an important role in… producing behaviors that support maternal care,” since we already know this statement to be true (as summarized in the following sentences.”

L56-58 – The point about “many species” doesn’t follow from the previous sentences, which mention only rats and mice.

L63 – As I mentioned in my previous review, I find the phrase “both non-lactating and lactating maternally-experienced females” confusing because it can be interpreted to mean that only the lactating females were maternally experienced. Maybe try rephrasing as “maternally experienced females that were either lactating or non-lactating”.

L68-69 – This sentence is very similar to the one at the end of the previous paragraph, which raises the question of why these two paragraphs are separate. Also, the first sentence of this paragraph (L59-60) seems to imply that the paragraph will focus on OXT receptors, which isn’t true of the second half of the paragraph. Consider removing the two sentences on receptors, since you didn’t quantify receptors in this study, and either reorganizing or consolidating the remainder of this paragraph and the previous paragraph.

L84 – Consider starting a new paragraph with the sentence beginning “A key social behavior…” Also, you might want to add something like “in the context of OXT” after “not been measured”.

L98 – Pultorak et al. 2017 is cited twice.

L104 – Change “have” to either “can have” or “has.”

L120 – Maybe change “responses” to “responsiveness” to make the statement a little broader? As is, it seems to refer only to the CRH and DEX challenges mentioned earlier in the same sentence.

L120-121 – The description of pair housing/family composition is out of order, since you don’t mention the actual pairing of animals until after the next sentence.

L124 – Again, it’s a little confusing to mention that pairs produced offspring before talking about when females were visibly pregnant (subsequent sentence).

L150 – Has IN treatment been shown to be less stressful than ICV treatment? If not, please indicate that this is presumed to be the case, rather than stating it as a fact.

L224 – The first part of this sentence is confusing. Do you mean that the videos were scored in random order (rather than the behaviors themselves)?

L294-297 – This seems backward. If scores from the reunion phase were subtracted from scores from the habituation phase, then positive difference scores indicate that habituation-phase scores were higher than reunion-phase scores.

L340-341 – The point about “both consistency and differences” is confusing, and the sentence overall isn’t particularly informative. I suggest that you delete it.

L347 – It’s not essential, but I think it would be helpful to specify that you’re referring to the number of maternal and pup calls (rather than some other measure). The same point applies to several other parts of the manuscript, including L408-411 and L440-441.

L396-403 – The first sentence of this paragraph indicates that the following analyses focus on effects of IN OXT, but the next two sentences sound like they’re describing within-subjects changes (or lack thereof) over time rather than effects of IN OXT. Please clarify.

L474 – As described in the methods, mothers had “multisensory pup experience,” although without touch, during the separation phase. Therefore, it would be more appropriate to emphasize the presence or absence of tactile stimuli in this sentence, rather than including this point only parenthetically. Same point for L480.

L477 – Delete “in” and “be.”

L561-563 – Which species do these references refer to?

7. PLOS authors have the option to publish the peer review history of their article (what does this mean?). If published, this will include your full peer review and any attached files.

Reviewer #1: No

Reviewer #2: No

Reviewer #3: No

---

## [Author Response · Author response to Decision Letter 1]

30 Mar 2021

Reviewer #1: All of my comments have been addressed in the revisions. The authors did a very nice job, this is a great contribution.

Reviewer #2: (No Response)

Reviewer #3: The authors were responsive to my previous comments, and the manuscript has improved. However, I have quite a few additional suggestions for increasing the clarify of the writing.

L12 – Mention saline controls.

L12—Amended

L19 - Not clear what “Maternal-pup USVs were correlated upon reunion” means – the number of USVs? Frequency? Timing? (Also, “Maternal-pup USVs” itself isn’t clear; try using something like “Mothers’ and pups’ USVs” instead.)

L19—Amended

L20 – “more positive change” is confusing. Change to something like “larger increase”?

L21—Amended

L24 – Change “single dose of” to “single treatment with”?

L28—Amended

L32 – Explain “pup whines.”

L38—Amended

L46-48 – It’s somewhat misleading to say that OXT “likely plays an important role in… producing behaviors that support maternal care,” since we already know this statement to be true (as summarized in the following sentences.”

L53—Removed likely

L56-58 – The point about “many species” doesn’t follow from the previous sentences, which mention only rats and mice.

L64—Amended

L63 – As I mentioned in my previous review, I find the phrase “both non-lactating and lactating maternally-experienced females” confusing because it can be interpreted to mean that only the lactating females were maternally experienced. Maybe try rephrasing as “maternally experienced females that were either lactating or non-lactating”.

L69-70—Amended

L68-69 – This sentence is very similar to the one at the end of the previous paragraph, which raises the question of why these two paragraphs are separate. Also, the first sentence of this paragraph (L59-60) seems to imply that the paragraph will focus on OXT receptors, which isn’t true of the second half of the paragraph. Consider removing the two sentences on receptors, since you didn’t quantify receptors in this study, and either reorganizing or consolidating the remainder of this paragraph and the previous paragraph.

L64-74—Amended

L84 – Consider starting a new paragraph with the sentence beginning “A key social behavior…” Also, you might want to add something like “in the context of OXT” after “not been measured”.

L104—Amended

L98 – Pultorak et al. 2017 is cited twice.

L117—Amended

L104 – Change “have” to either “can have” or “has.”

L123—Amended

L120 – Maybe change “responses” to “responsiveness” to make the statement a little broader? As is, it seems to refer only to the CRH and DEX challenges mentioned earlier in the same sentence.

L104—Amended

L120-121 – The description of pair housing/family composition is out of order, since you don’t mention the actual pairing of animals until after the next sentence.

L140—Amended

L124 – Again, it’s a little confusing to mention that pairs produced offspring before talking about when females were visibly pregnant (subsequent sentence).

L144-145—Amended

L150 – Has IN treatment been shown to be less stressful than ICV treatment? If not, please indicate that this is presumed to be the case, rather than stating it as a fact.

L172—Amended

L224 – The first part of this sentence is confusing. Do you mean that the videos were scored in random order (rather than the behaviors themselves)?

L246—Amended

L294-297 – This seems backward. If scores from the reunion phase were subtracted from scores from the habituation phase, then positive difference scores indicate that habituation-phase scores were higher than reunion-phase scores.

L318—Amended

L340-341 – The point about “both consistency and differences” is confusing, and the sentence overall isn’t particularly informative. I suggest that you delete it.

L364-365—Amended

L347 – It’s not essential, but I think it would be helpful to specify that you’re referring to the number of maternal and pup calls (rather than some other measure). The same point applies to several other parts of the manuscript, including L408-411 and L440-441.

L371, 436—Amended

L396-403 – The first sentence of this paragraph indicates that the following analyses focus on effects of IN OXT, but the next two sentences sound like they’re describing within-subjects changes (or lack thereof) over time rather than effects of IN OXT. Please clarify.

L426—Amended

L474 – As described in the methods, mothers had “multisensory pup experience,” although without touch, during the separation phase. Therefore, it would be more appropriate to emphasize the presence or absence of tactile stimuli in this sentence, rather than including this point only parenthetically. Same point for L480.

L502. 508—Amended

L477 – Delete “in” and “be.”

L505—Amended

L561-563 – Which species do these references refer to?

L595-596—Amended

---

## [Editor Report · Decision Letter 2]

6 Apr 2021

An acute dose of intranasal oxytocin rapidly increases maternal communication and maintains maternal care in primiparous postpartum California mice

PONE-D-20-37165R2

Dear Dr. Guoynes,

We’re pleased to inform you that your manuscript has been judged scientifically suitable for publication and will be formally accepted for publication once it meets all outstanding technical requirements.

Kind regards,

Cheryl S. Rosenfeld, DVM, PhD

Section Editor

PLOS ONE
---

## [Editor Report · Acceptance letter]

12 Apr 2021

PONE-D-20-37165R2 

An acute dose of intranasal oxytocin rapidly increases maternal communication and maintains maternal care in primiparous postpartum California mice 

Dear Dr. Guoynes:

I'm pleased to inform you that your manuscript has been deemed suitable for publication in PLOS ONE. Congratulations! Your manuscript is now with our production department. 

Kind regards, 

on behalf of

Dr. Cheryl S. Rosenfeld 

Section Editor

PLOS ONE